# Prevention of congenital toxoplasmosis in France using prenatal screening: A decision-analytic economic model

**Larry Sawers**[1], **Martine Wallon**[2,3], **Laurent Mandelbrot**[4,5,6], **Isabelle Villena**[7,8], **Eileen Stillwaggon**[9], **François Kieffer**[10]*

**1** Department of Economics, American University, Washington, D.C., United States of America, **2** Department of Parasitology and Medical Mycology, Hospices Civils de Lyon, Lyon, France, **3** Walking Team, Centre for Research in Neuroscience, Lyon, Bron, France, **4** Obstetrics and Gynecology Department, Hôpital Louis Mourier, Assistance Publique-Hôpitaux de Paris, Colombes, France, **5** Université de Paris, Paris, France, **6** INSERM, IAME, UMR 1137, Paris, France, **7** Department of Parasitology and Medical Mycology, National Reference Centre on Toxoplasmosis, Hôpital Reims, Reims, France, **8** Team EA 7510, SFR CAP-SANTE, University of Reims Champagne Ardenne, Reims, France, **9** National School of Tropical Medicine, Baylor College of Medicine, Houston, Texas, United States of America, **10** Department of Neonatology, Hôpital Armand Trousseau, Assistance Publique-Hôpitaux de Paris, Paris, France

* francois.kieffer@aphp.fr










**Data Availability Statement:** All relevant data are within the paper and Supporting Information files.

**Funding:** The author(s) received no specific funding for this work.

## Abstract

### Background

*Toxoplasma gondii* is one of the world's most common parasites. Primary infection of the mother during pregnancy can lead to transmission to the fetus with risks of brain and eye lesions, which may cause lifelong disabilities. France instituted a national program based on monthly retesting of susceptible pregnant women to reduce the number of severe cases through prompt antenatal and postnatal treatment and follow-up.

### Objective

To evaluate the ability of the French prenatal retesting program to reduce the lifetime costs of congenital toxoplasmosis.

### Methods

We measured and then compared the costs and benefits of screening vs. not screening using decision-tree modelling. It included direct and indirect costs to society of treatment and care, and the lifetime lost earnings of children and caregivers. A probabilistic sensitivity analysis was carried out.

### Findings

Total lifetime costs per live born child identified as congenitally infected were estimated to be €444 for those identified through prenatal screening vs €656 for those who were not screened. Estimates were robust to changes in all costs of diagnosis, treatment, and sequelae.

**Competing interests:** The authors have declared that no competing interests exist.

## Interpretation

Screening for the prevention of the congenital *T. gondii* infection in France is cost saving at €212 per birth. Compared with no screening, screening every pregnant woman in France for toxoplasmosis in 2020 would have saved the country €148 million in addition to reducing or eliminating the devastating physical and emotional suffering caused by *T. gondii*. Our findings reinforce the conclusions of other decision-analytic modelling of prenatal toxoplasmosis screening.

## Introduction

*Toxoplasma gondii* (*T. gondii*) is a protozoan parasite infecting approximately a third of all humans [1], with substantial differences in prevalence across the globe. Sources of human infection include the ingestion of cysts in raw or undercooked contaminated meat, or of oocysts present in cat feces or contaminated vegetables, fruits, soil, and water [1]. In case of primary infection with *T. gondii* during pregnancy, the risk of transmission is very low in the first trimester (2%-5%) and reaches 70% in the final weeks of pregnancy [2]. Spontaneous abortion may occur in case of early fetal infection. The probability of mild or subclinical disease is highest in infection acquired in the later stages of gestation [1, 3–5]. Sequelae of congenital toxoplasmosis (CT) include cerebral calcifications, hydrocephaly, cognitive and motor sequelae, retinochoroiditis, and visual and hearing impairment.

Prenatal screening programs are operated at a national scale in France, Austria, and Slovenia. Since 1985, all susceptible pregnant women in France are to be identified at their first prenatal visit, and since 1992, they are to be retested monthly until delivery. The objectives are to allow prompt treatment in case of seroconversion to prevent mother-to-child transmission and treatment of infected fetuses to reduce the likelihood and severity of injury [2, 6–14].

In recent decades, *T. gondii* seroprevalence has decreased sharply in industrialized countries [15–17]. In France, the seroprevalence in pregnant women fell from 54% in 1995 to 31.3% in 2016 [18], and the incidence of seroconversions during pregnancy decreased from 5.4 per 1,000 at-risk pregnancies in 1995 to 3.1 per 1,000 in 2016 with a prediction of 1.6 per 1,000 susceptible women by 2020 [19]. The cost of a screening program increases with the number of susceptible pregnant women to retest monthly. The declining seroprevalence challenges prenatal screening programs. Paradoxically, the success of maternal prenatal screening for *T. gondii* in France and Austria in reducing the number of children with disabling forms of CT has also undermined support in those countries for maintaining expensive prenatal screening programs [11, 20]. Two cost-benefit analyses (CBA) found prenatal retesting for toxoplasmosis to be cost saving, one with the hypothesis of applying the French screening protocol in the USA [21], and the other examining the Austrian screening protocol [20]. A cost-effectiveness study in 2019 compared neonatal to prenatal screening and found the French prenatal screening program to be cost effective [22].

To evaluate the ability of the monthly retesting program to reduce the lifetime costs of CT in the current epidemiological context, we performed a cost-benefit analysis (CBA) of prenatal screening in France. Our study was designed to facilitate comparison with two previous CBAs of toxoplasmosis screening.

## Methods

### Description of French toxoplasmosis screening and management protocol

Our modelling of maternal screening for toxoplasmosis is based on the protocol described by the National College of French Obstetricians and Gynecologists [11]. The first step in the

screening process is testing pregnant women without previous documented immunization for anti-*T*. *gondii* IgG and IgM in the first trimester. Women whose serology is negative are informed about prevention of toxoplasmosis and are retested monthly until delivery [23]. To limit the passage of the parasite from mother to fetus, treatment is initiated as soon as a maternal infection is detected. Treatment includes spiramycin, or, after 14 weeks of gestation, pyrimethamine-sulfadiazine combination. Amniocentesis is offered at least 4 weeks from the presumed date of maternal infection and from 18 weeks of gestation. If the result of the PCR on the amniotic fluid is negative, spiramycin is continued until delivery. As the specificity of PCR on amniotic fluid is 100%, a positive result indicates that the fetus has CT and pyrimethamine-sulfadiazine is prescribed until delivery [24]. Prenatal ultrasound is performed monthly if the PCR result is negative and bimonthly if positive. In the event of severe fetal abnormalities, termination of pregnancy may be performed at the patient's request after approval by a multidisciplinary prenatal diagnosis centre. All newborns undergo assessment at birth that include cerebral ultrasonography, an ocular examination, and testing of neonatal blood for anti-*T*. *gondii* IgM, IgA, and IgG antibodies. Infants with proven CT are treated with pyrimethamine and sulfadiazine (or sulfadoxine) for one year after birth and monitored in the long-term for neurological and ophthalmological complications. In the other children, the absence of infection is demonstrated by monitoring the clearance of maternal IgG before age one.

## Modeling

We assumed full compliance with all procedures in the screening scenario and that all medications were well tolerated without serious adverse effects. In the no-screening scenario, we assumed that no subjects would be identified through individual screening, through testing in the context of maternal clinical signs of toxoplasmosis, or in the context of fetal ultrasound anomalies. We also assumed that children who are identified with CT in the no-screening scenario would still be treated for 12 months with no side effects just as in the screening scenario, even in cases of more severe or subclinical infection, and even if diagnosed several months or years after birth. Monitoring of infected children was assumed the same in both the screening and no-screening scenarios.

We used TreeAge Pro Suite 2021 software (TreeAge Software, Inc., Williamstown, MA, US) to construct a decision-analytic model. Our CBA is built on two decision-analytic models, the retesting program currently operated in Austria, and a hypothetical program in the United States adapted from the French program [21]. We adapted those decision trees to use as templates for our modelling. The most important difference between the Austrian and French protocols is that the former has 4 serological tests 8 weeks apart while the latter has 8 serological tests 4 weeks apart. The protocol of management used in France has changed in important ways since 2011 when the US study was carried out and our decision tree reflects these improvements, particularly regarding prenatal diagnosis and treatments [25]. Images of the decision tree appear in the S1 File. They show the probabilities of all possible outcomes due to CT (green circles at chance nodes) and the costs associated with each outcome (red triangles at terminal nodes). Each outcome has a conditional probability equal to the sum of the probabilities along each branch. The formulas at the terminal nodes for each outcome provide the current cost of testing, surveillance, and medications, and costs of all direct injury costs (such as remedial education) and all indirect injury costs (such as lost earnings due to impairment).

## Clinical probabilities

Probabilities of clinical outcomes were gathered from published estimates in both the context of no-screening (Table 1) and screening (Table 2). Examples include the probability of primary

**Table 1. Probabilities with no-screening option.**

| Variable | Probabilities (range) | Reference |
|---|---|---|
| Primary infection in pregnancy | 0.0016 | [7, 8, 18] |
| CT | 0.39 (0.31–0.47) | [26] |
| Fetal death due to CT | 0.0135 | [8] |
| No CT | 0.06 | [26] |
| Visual injury | 0.48 | [26–29] |
| Of which Mild | 0.09 | [26–29] |
| Visual and cognitive injury | 0.45 (0.40–0.55) | [5, 26, 28] |
| Of which Mild | 0.39 (0.33–0.45) | [5, 26, 28] |
| Visual, cognitive, and hearing injury | 0.01 | [5, 26, 28] |


infection during pregnancy and the probabilities of pediatric clinical long-term outcomes due to congenital toxoplasmosis (CT) such as mild visual impairment. Table 1 shows the probabilities in the decision tree with the no-screening option while Table 2 shows the probabilities in the decision tree with the screening option. The order of the variables in the tables follows the structure of the decision tree. At the top of the table are located the variables of the beginning of the tree. At the bottom of the table are the variables at the end of each branch. At each intermediate node, the sum of the probabilities equals 1. The order of the variables in the table follows the structure of the tree. At the top of the table are located the variables of the beginning of the tree. At the bottom of the table are the variables at the end of each branch.

**Table 2. Probabilities with screening option.**

| | date of maternal primary infection | | | | | | | | | |
|---|---|---|---|---|---|---|---|---|---|---|
| **Probabilities** | **12 Weeks** | **16 Weeks** | **20 Weeks** | **24 Weeks** | **28 Weeks** | **32 Weeks** | **36 Weeks*** | **Newborn *** | **Postnatal *** | **References** |
| Maternal IgG (+) (Seroprevalence) | 0.313 | | | | | | | | | [18] |
| IgG(+) IgM(+) Primary Maternal Infection | 0.0016 | | | | | | | | | [18] |
| Confirmation test (+) | 0.9 | | | | | | | | | [1] |
| Fetal death due to CT | 0.02 | 0.0025 | 0 | 0 | 0 | 0 | 0 | 0 | 0 | [30] |
| Fetal loss from amniocentesis | 0.003 (0.0011–0.0049) | 0.003 | 0.003 | 0.003 | 0 | 0 | | | | [31] |
| Amniocentesis (-) | 0.90 | 0.873 | 0.785 | 0.638 | 0.514 | 0.450 | | | | [8, 24, 32] |
| False negative of amniocentesis (CT if amniocentesis (-)) | 0.05 | 0.05 | 0.05 | 0.05 | 0.10 | 0.15 | | | | [8, 24, 32] |
| Asymptomatic CT if amniocentesis (-) | 0.70 | 0.70 | 0.75 | 0.775 | 0.79 | 0.79 | 0.795 | 0.795 | 0.795 | [8, 24, 32] |
| Visual injury if CT with amniocentesis (-) | 0.20 | 0.20 | 0.20 | 0.20 | 0.20 | 0.20 | 0.20 | 0.20 | 0.20 | [2, 12, 14, 33, 34] |
| Visual and cognitive injury if CT with amniocentesis (-) | 0.10 | 0.10 | 0.05 | 0.025 | 0.01 | 0.01 | 0.005 | 0.005 | 0.005 | [2, 12, 14, 33, 34] |
| Asymptomatic CT if amniocentesis (+) | 0.50 | 0.5 | 0.70 | 0.75 | 0.795 | 0.795 | | | | [2, 12, 14, 33, 34] |
| Visual injury if CT with amniocentesis (+) | 0.30 | 0.30 | 0.25 | 0.20 | 0.20 | 0.20 | | | | [2, 12, 14, 33, 34] |
| Visual and cognitive injury if CT with amniocentesis (+) | 0.20 | 0.20 | 0.05 | 0.05 | 0.005 | 0.005 | | | | [2, 12, 14, 33, 34] |

*Probability without amniocentesis and with confirmatory testing


## Economic costs

Two types of costs are considered in the analysis, The first is the cost of treatment, which includes medication and testing during the mother's pregnancy and in the year following her child's birth. These are presented in Table 3. The second type of costs considered in the analysis are injury costs. They include remediation (for example, the costs of hearing aids for those with hearing damaged by CT, costs of remedial education for children with mild or severe cognitive impairment due to CT, or lifetime earnings loss due to disability caused by CT.

Some of these costs such as medications and tests are purchased and used within a few months or a single year. Other costs persist for years, for example lifetime earnings loss for patients who are never able to work (measured from average age of those first entering the paid labor force to the average age at retirement). Health economists in the United States routinely assume that a euro received tomorrow is worth 3% less (that is, discounted annually by 3%) than a euro received today. Other costs of other injuries may last only a few years. An example is the cost of remedial education when the child is young or the lost wages of a parent who stays at home to care for an injured child. A description of how these costs were measured is presented in S1 File.

Our aim is to measure these costs from the perspective of society as a whole regardless of which person, institution, or medical insurance system paid for the treatment or injury costs. In the interest of simplicity, we assume 100% compliance with the protocol that we are testing. For example, we assume that if amniocentesis is medically recommended, that the mother complies with the recommendation. We further assume there are no side effects to medications or procedures. Tables 3 and 4 summarize costs that were included in the decision-tree analysis and the sources used to measure those costs. Table 3 reports the cost of spiramycin and pyrimethamine and sulfonamides, and the costs of tests performed to diagnose infections and complications and to monitor side effects of treatment. Table 4 summarizes costs that arise from injuries and impairments to persons with CT, their families, and the economy

**Table 3. Test and medication costs.**

| Variable Name in Tree | Descriptions | Current Value in 03/2021 | Source (code) |
|---|---|---|---|
| MatIgGMTest | Maternal IgG + IgM test | €10.8 | a (40B) |
| Amnio | Amniocentesis with PCR on amniotic fluid | €40.5 | a (150B) |
| MatCBC | CBC during maternal treatment | €6.75 | a (25B) |
| ObstetricUltra | Prenatal diagnostic ultrasound | €100.2 | b (JQQMO18) |
| RxNegPCR | Maternal treatment with negative PCR on amniotic fluid (spiramycin for 4 months) [23] | €207.72 | c |
| RxPosPCR | Maternal treatment with positive PCR on amniotic fluid (spiramycin for one month, then pyrimethamine, sulfadiazine and folinic acid for 3 months) [23] | €195.19 | c |
| InfIgGMTest | Infant IgG + IgM test | €10.8 | a (40B) |
| WestBlotTest | Western Blot test (comparison test of IgG and IgM profiles of mother and child) | €86.4 | a (320B) |
| PedCranialUltra | Cerebral ultrasound | €37.8 | b (AAQM002) |
| Fundus | Funduscopy | €48.36 | b (BGQP004) |
| PedRx | Pediatric treatment 12 months | €3124.49 | C |
| PedCBC | CBC during pediatric treatment | €6.75 | a (25B) |

*Source*: a. www.codage.ext.cnamts.fr B value €0.27

b. https://www.aideaucodage.fr/ccam

c. https://www.vidal.fr/medicaments/


**Table 4. Costs due to impairments.**

| Impairment name in tree | Description of Impairments | Cost of impairment |
|---|---|---|
| CognitiveMild | Treatment for mild cognitive impairment | €54,124 |
| CognitiveSevere | Treatment for severe cognitive impairment | €406,634 |
| HearingMild | Treatment for mild hearing impairment | €22,343 |
| SpecEdBlind | Special school for severe visual impairment | €91,993 |
| SpecEdCognitiveMild | Special school for mild cognitive impairment | €74,967 |
| SpecEdCognitiveSevere | Special school for severe cognitive impairment | €747,175 |
| VisualMild | Treatment for mild visual impairment | €272,684 |
| ChildEarnLoss | Earnings loss for severe cognitive impairment | €564,858 |
| ParentEarnLoss | Earnings loss of caregiver | €28,262 |
| VisualSevere | Income loss + non-medical costs of severe visual impairment | €590,374 |
| VSL | Value of a statistical life | €5.58 million |

*Source*: (S1 File): Measuring costs of impairment at terminal nodes in decision tree


regardless of who pays. Examples include care for patients with cognitive and visual impairment and earnings loss of patients with CT and of their caregivers. Estimates of costs are derived from the literature for France (adjusted to the price level in 2020).

The decision tree we used to calculate the cost savings from maternal screening for toxoplasmosis can be found in S1 File. The costs and the probabilities of incurring those costs are shown along each branch of the tree. The image of the entire decision tree (S1 Fig) gives the viewer a sense of the complexity of the estimations produced by the decision tree analysis, but the numbers and words on each branch are too small to decypher without magnification. Accordingly, in the S1 File we also include separate images of the 9 major branches of the decision tree that together form the decision tree (S2–S10 Figs). The last image shows the simplified tree after 4 of 5 identical branches of the tree were omitted (S11 Fig).

We used an incremental tornado diagram (see Fig 1) to test the robustness of our results to variations in costs. The tornado diagram is an array of one-way sensitivity tests ordered by the ability of each variable to affect cost savings produced by prenatal screening. We performed a sensitivity analysis using an Incremental Tornado Diagram varying all costs by ±20%. The model was based on previously obtained data from the literature, so no informed consent was possible or required.

The x-axis of the diagram shows cost saving per birth produced by prenatal screening. The horizontal bars show the variation in the expected per-birth value of cost saving from screening resulting from changing the range of values for each cost parameter by plus 20% and minus 20%.

## Results

S1 File. show the decision tree after calculation of the no-screening option and the screening option. Without screening, the lifetime societal costs of CT sequelae plus treatment of toxoplasmosis for twelve months would have been €656 per birth in 2020 while the costs with screening were €444. Thus, screening saved €212 per birth. There were an estimated 700,000 births in France in 2020 [35]. Assuming every pregnant mother was screened according to protocol in 2020, the cost saving would have approached €148 million in France that year.

Of the costs examined in the incremental tornado diagram (Fig 1), variations in the probability of maternal primary infection had the greatest effect on cost saving from prenatal screening. Varying it by plus or minus 20% changed cost saving by 30.5%. The cost of treatment for

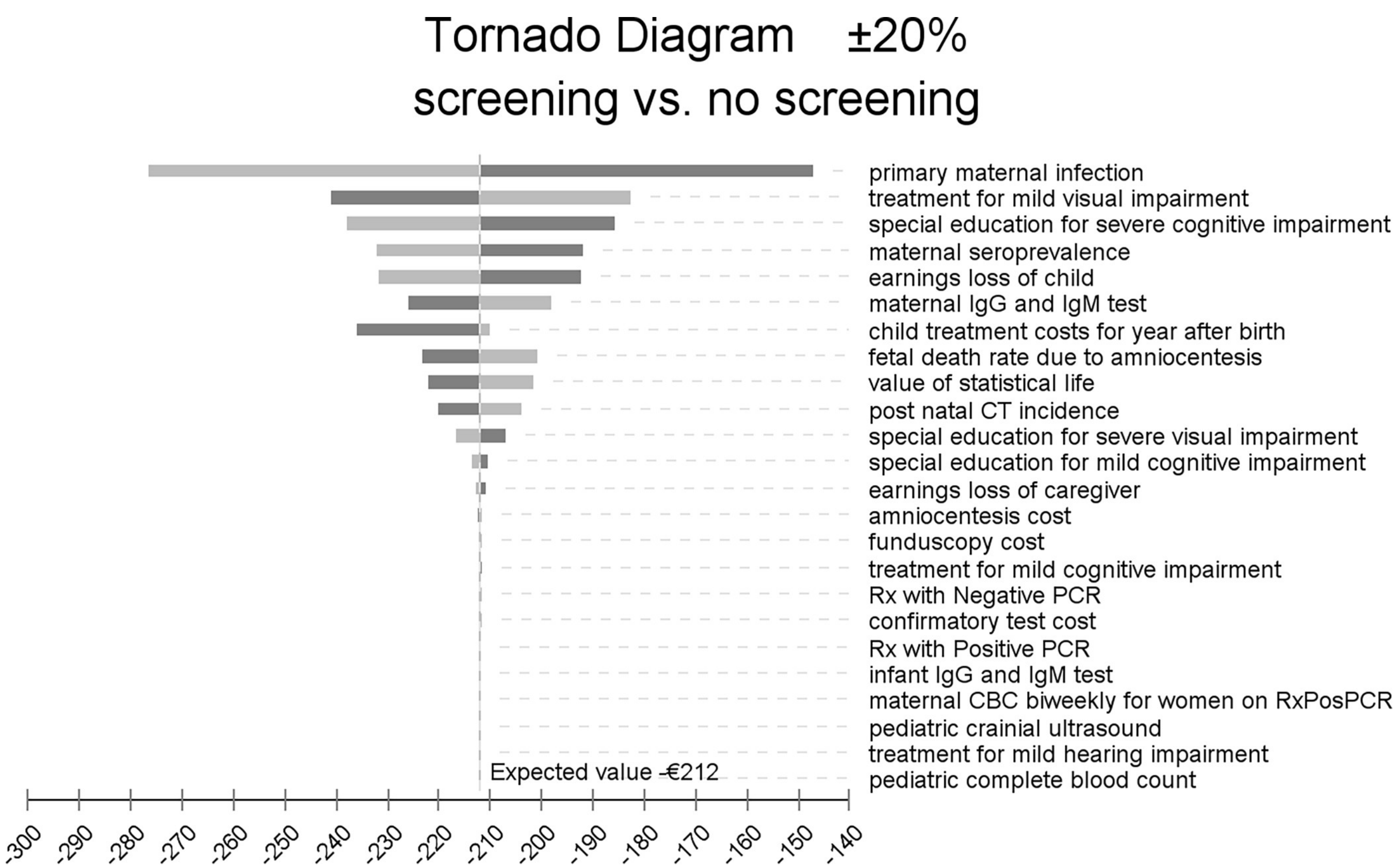


**Fig 1. Tornado diagram.**


mild visual impairment was the next most important variable. Changing it by ±20% led to a 14% change in estimated cost saving produced by prenatal screening. The third most important variable in affecting estimated cost saving from prenatal screening was the cost of special education for those with severe cognitive impairment. A ±20% change in those costs led to a 12% change in estimated cost saving. Increasing or decreasing all other variables by 20% led to estimated cost saving changing by less than 10%. Most of the variables had a trivial impact on cost saving from screening. For two-thirds of the 24 variables in the tornado analysis with the least impact, a 20% increase or decrease in value led to a change in cost saving from screening by less than 5%.

The one-way sensitivity analysis in Fig 2 examines changes in the incidence of primary maternal infection. This variable in the tornado diagram has the greatest impact on reducing screening costs and is the only parameter that can change the optimal strategy. The breakeven point shown in Fig 2 is 0.0005504 (5.504 per 10,000), indicating that the incidence of primary maternal infection would have to fall by nearly two thirds for a no-screening strategy to become less costly than the actual screening strategy. No other variable had a breakeven point.

## Discussion

To reduce the lifetime consequences of congenital Toxoplasma infection, two preventive strategies are considered, prenatal and neonatal. The prenatal strategy combines education and serological testing of susceptible pregnant women, with three objectives: to avoid maternal infection, to recognize infection promptly, and to detect and treat before birth any congenital

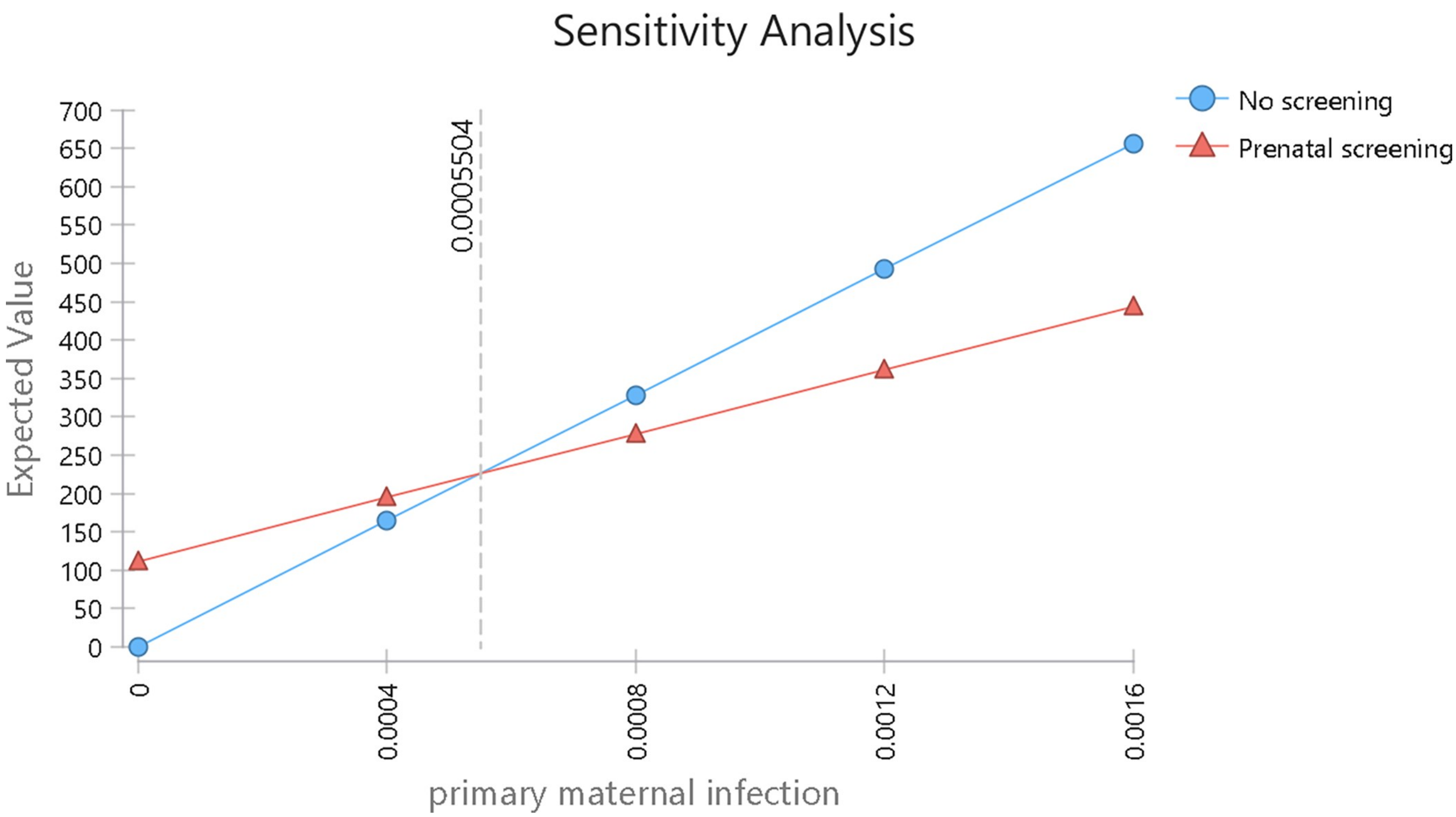


**Fig 2. Sensitivity analysis of the expected value of prenatal screening for toxoplasmosis vs no screening, according to the annual incidence of primary maternal infection.**


infection. The prenatal approach relies on the hypotheses, sustained by indirect evidence, that early maternal treatment reduces the risk of mother-to-child transmission [2], and that congenital infection treated prenatally is associated with a lower risk of severe lesions [13]. France has organized this prenatal strategy at a national level, including fully reimbursed retesting, every month, which makes it the most comprehensive program worldwide.

The aim of our study was to estimate whether this program is cost saving despite the declining seroprevalence and incidence of maternal and fetal infections. The model was based on population-based published outcomes and direct and indirect costs for society.

Our main finding is that monthly retesting is cost saving for French society since it reduces the number of severe cases of CT. We estimate that without prenatal screening, the cost of treating CT and dealing with long term functional consequences would be €656 per birth. Prenatal screening reduced the cost to €444, generating savings of €212 per birth. If no pregnant woman in France was screened for toxoplasmosis in 2020, the societal costs would have been €460 million.

Our findings reinforce the conclusions of the cost-effectiveness analysis in 2019 that found prenatal screening according to the French protocol was cost effective compared with neonatal screening, at endpoints of one year and 15 years [22]. Our findings also agree with the two previous CBA of maternal screening for toxoplasmosis, despite differences in management of prenatal infections, retesting schedules, prevalence, and injury costs. The analysis in the low prevalence context of the United States in 2011 estimated that the screening program was cost saving and a sensitivity analysis estimated that it would still be the case for even lower prevalence [21]. The CBA of maternal toxoplasmosis screening in Austria found that prenatal screening for toxoplasmosis led to savings of €323 per birth in 2016 [20]. Adjusting for euro

inflation between 2016 and 2020 and taking into account the lower estimates of prevalence in Austria narrowed the difference between the French and Austrian studies. The CBA in the US in 2011 and in France in 2021 assumed similar screening protocols and found similar level of costs for the screening option. After adjusting for inflation between 2011 and 2020, the cost of screening in the US of $390 was only 6.6% higher than the cost of the screening option in the French 2021 CBA analysis. The lower medical costs in France in 2020 than in the US in 2011 is partly explained by the 5-times higher cost of amniocentesis in the US in 2011 compared to France in 2021. Another part of the explanation reflects the technological improvements in diagnosis of CT in the decade after 2011. For example, the cost of obstetric ultrasounds (€100 each) in the French tree do not appear in the US tree. The medical regimen also changed between 2011 and 2020. Prenatal treatment with sulfadiazine and pyrimethamine is now given in second trimester maternal seroconversions as soon as the diagnosis is made and not only in case of positive amniocentesis and requires weekly CBC [11, 22]. Finally, the cost of pediatric treatment over one year for a child with CT was $210 in the US in 2011 and was €3,124 in France in 2021.

Costs for the no-screening scenario in the US CBA were greater by 75% ($1010 in 2011, equivalent to €1429 in 2021 after adjusting for inflation) than in our study (€656). This difference partially reflects the higher costs of medication but mostly results from the dramatically higher costs of caring for CT-related injuries, including remedial education, custodial care, hearing aids, eyeglasses, earnings loss by caregivers and those disabled by toxoplasmosis in the US [36, 37]. The arbitrarily higher estimate of the value of statistical life in the US combined with the higher probability of fetal death in the US CBA also help explain the difference in costs in the no-screening scenario. In summary, our CBA and the US and Austrian CBAs produce mostly similar conclusions. Dissimilar results from the three studies are explained by the substantial differences in the protocol used in each study and some costs of medicines, medical procedures, and injury costs.

A second important finding was the robust results for most parameters analysed in the sensitivity analysis. It includes prevalence of infection, which is a key finding at a time of falling prevalence in industrialized countries. Falling prevalence is also increasingly reported in developing countries in the few recent studies including valid comparisons over time [1].

Due to their common structure, our models shared several characteristics with the two other CBA models. The first is that probabilities of clinical outcomes were based on population-based reports, in the context of screening or no screening conversely and not on pre-set estimates of treatment efficacy. This allowed estimating the impact of prompt diagnosis and treatment interventions on reducing the number of severe cases of CT despite the lack of randomized controlled trials that compared screening to no screening. Experimental, parasitological, and clinical data indicate that prompt initiation of anti-parasitic treatment following maternal infection reduces the risk of placental transmission. Several recent studies from Europe and South America that adjust for the gestational age at the time of maternal infection (the major risk factor for transmission) observed lower rates of transmission with pyrimethamine-sulfadiazine than other or no prophylaxis [38]. A single RCT found a trend towards less transmission with pyrimethamine-sulfadiazine than spiramycin (18.5% *vs*. 30%, p = 0.147). This association was strengthened when the treatment was started within 3 weeks of seroconversion. The incidence of fetal cerebral ultrasound signs was significantly lower in the pyrimethamine-sulfadiazine group [39]. The benefits of anti-parasitic therapy for fetuses or neonates with CT are also strongly suggested by large observational studies, which show that prompt initiation of treatment with pyrimethamine and sulphonamides is associated with a decreased incidence of cerebral signs and symptoms as well as retinochoroidal lesions [12, 14]. However, no valid quantitative estimates are available to quantify the risk reduction from

treatment. Binquet *et al*. made the choice to model the impact of treatment based on several assumptions of efficacy [22].

For the sake of simplicity and comparability with the two previous CBA studies, we also made the choice not to include the possibility for acute maternal infection to be detected by clinical signs; these are present in 20 to 30% of cases in immunocompetent adults but are non-specific and often identified only after seroconversion is diagnosed [1]. We also did not include the cost of the individual screening, which is increasingly performed worldwide at the initiative of patients at individual clinical practices or maternity hospitals or at the request of patients. These aggregated costs might have reduced the difference in costs and benefits between screening and no screening. There were, however, not found by Binquet *et al*. to modify their conclusion regarding the cost effectiveness of the monthly prenatal screening [22]. Similarly, we assumed 100% compliance with the screening, diagnosis, treatment procedures, and lack of side effects from treatment. There is no precise estimate of suboptimal adherence in the real world. In a study in one region of France in 2009, a quarter of the participants had their first test beyond the first trimester of pregnancy, 80% of participants had at least one between-test interval exceeding 35 days, and 60% of participants completed fewer than the recommended seven tests [40]. Non-compliance probably led to overestimating both the cost and benefits of screening in proportions that cannot be precisely estimated. Varying compliance from 50 to 100% however did not significantly affect the conclusion by Binquet *et al*. [22].

Also, we did not include indirect spillover benefits of serological retesting and health education on the overall outcomes of pregnancy, which could have increased the benefits associated with screening. Awareness of preventing infection through improved hand hygiene and food preparation could be reinforced by performing repeated blood tests. Finally, we did not model the psychological consequences of prenatal screening. Repeated monthly tests, amniocentesis, or uncertainties in the prognosis for fetuses with CT can be responsible for significant anxiety. Conversely, the psychological benefits of prenatal screening are also real since rapid treatment and precise information can be reassuring [41]. Costs of training clinicians and biologists should not be underestimated when implementing a screening program and would need to be included in a further stage of planning.

The simple structure of our tree will facilitate monitoring the impact of any additional preventive interventions or changes in the epidemiology of *T. gondii*. They will also help measuring the saving produced by using the point-of-care tests that have been validated as excellent and inexpensive and would reduce the logistical constraints of the French universal retesting program [42]. Our model could also be used as a simple decision-making tool for helping policy makers worldwide in simulating the benefits that implementing prenatal screening might have in their settings, according to local epidemiological and clinical findings, and costs. This approach would be of upmost interest in areas where Toxoplasma gondii is severe due to the presence of more virulent strains. It could also be used to help confirm the decision of not to screen made in several countries.

The finding that maternal screening is cost saving in both Austria and France–and would be in the US if implemented there–serves to reinforce confidence in ongoing prenatal toxoplasmosis screening programs. This also encourages efforts to implement prenatal screening programs in the US and elsewhere, particularly in countries where the incidence of toxoplasmosis is high, or the strains virulent, and where the socio-economic level allows access to prenatal diagnosis. Even in a country where prevalence is low and falling, prenatal screening may be needed, since we found that the breakeven point for prenatal screening was reached with a primary maternal infection rate much lower than the current French rate.

## Conclusion

This analysis shows that prenatal screening and treatment for *T*. *gondii* infection following the French protocol result in substantial cost saving. Our results are robust to wide variations in parameter values, and they indicate that prenatal toxoplasmosis screening may be beneficial even in countries with low prevalence.

## Supporting information

**S1 File. Measuring the costs of impairment used at terminal nodes in the decision tree.**
(DOCX)

**S1 Fig. Image of entire decision tree with the 9 major branches: These 9 branches together show the probabilities of all possible outcomes due to congenital toxoplasmosis (green circles at chance nodes) and the costs associated with each outcome (red triangles at terminal nodes).** Each outcome has a conditional probability equal to the sum of the probabilities along each branch. The formulas at the terminal nodes for each outcome provide the current cost of testing, surveillance and medications, and costs of all direct injury costs (such as remedial education) and all indirect injury costs (such as lost earnings due to impairment) as shown inS1 Fig. Since the decision tree reproduced into a single image proved to be very difficult to read, we segmented the tree into its 9 main branches (S2–S10 Figs) and a simplified version of decision tree (S11 Fig).
(TIF)

**S2 Fig. No screening.**
(TIF)

**S3 Fig. Screening and maternal seroconversion before 12 weeks.**
(TIF)

**S4 Fig. Screening and maternal seroconversion between 12 and 16 weeks.**
(TIF)

**S5 Fig. Screening and maternal seroconversion between 16 and 20 weeks.**
(TIF)

**S6 Fig. Screening and maternal seroconversion between 20 and 24 weeks.**
(TIF)

**S7 Fig. Screening and maternal seroconversion between 24 and 28 weeks.**
(TIF)

**S8 Fig. Screening and maternal seroconversion between 28 and 32 weeks.**
(TIF)

**S9 Fig. Screening and maternal seroconversion between 32 and 36 weeks.**
(TIF)

**S10 Fig. Screening and maternal seroconversion after 36 weeks and no maternal seroconversion.**
(TIF)

**S11 Fig. Simplified version of decision tree, omiting 4 of 5 identical tree branches.**
(TIF)

## Author Contributions

**Conceptualization:** Larry Sawers, Martine Wallon, Eileen Stillwaggon, François Kieffer.

**Formal analysis:** Larry Sawers.

**Investigation:** Larry Sawers, Martine Wallon, François Kieffer.

**Methodology:** Larry Sawers, Martine Wallon, Eileen Stillwaggon, François Kieffer.

**Supervision:** Larry Sawers.

**Writing – original draft:** Larry Sawers, Martine Wallon, François Kieffer.

**Writing – review & editing:** Larry Sawers, Martine Wallon, Laurent Mandelbrot, Isabelle Villena, François Kieffer.

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
