## [Decision Letter · Decision Letter 0]

10 Feb 2022

PONE-D-21-31899Prevention of congenital toxoplasmosis in France using prenatal screening: A decision-analytic economic modelPLOS ONE

Dear Dr. Sawers,

Thank you for submitting your manuscript to PLOS ONE. After careful consideration, we feel that it has merit but does not fully meet PLOS ONE’s publication criteria as it currently stands. Therefore, we invite you to submit a revised version of the manuscript that addresses the points raised during the review process.

 Thank you for submitting this interesting article. Please take note of input from scientific reviewers, particularly methodologic details from reviewer #4.  The full reviewer input is attached separately.

We look forward to receiving your revised manuscript.

Kind regards,

Jodie Dionne-Odom, MD

Academic Editor

PLOS ONE

Journal Requirements:

No authors have competing interests

Reviewers' comments:

Reviewer's Responses to Questions

**Comments to the Author**

1. Is the manuscript technically sound, and do the data support the conclusions?

Reviewer #1: Partly

Reviewer #2: Yes

Reviewer #3: Yes

Reviewer #4: Partly

2. Has the statistical analysis been performed appropriately and rigorously? 

Reviewer #1: Yes

Reviewer #2: Yes

Reviewer #3: Yes

Reviewer #4: Yes

3. Have the authors made all data underlying the findings in their manuscript fully available?

Reviewer #1: No

Reviewer #2: Yes

Reviewer #3: Yes

Reviewer #4: Yes

4. Is the manuscript presented in an intelligible fashion and written in standard English?

Reviewer #1: Yes

Reviewer #2: Yes

Reviewer #3: Yes

Reviewer #4: No

5. Review Comments to the Author

Reviewer #1: This is an excellent paper. The topic is highly relevant, the methodology is appropriate, it is well-written. I support publishing under few conditions.

Firstly, your results strongly depend on the costs. Even if the tornado diagram (+/- 20 %) indicates that the intervention remains cost-saving, there is still a major question about the costs. I would like to know much more about how you calculated the costs. At least I would like to see a detailed analysis of the underlying literature. How did you calculate the intervention cost? What part of the life-long cost are direct and indirect? How were indirect cost / opportunity cost calculated? and if you take them from literature: are they really comparable? What interest rate did you include? In other words: write much more about the costs.

Secondly, I had severe problems to read the decision-tree. It might be worthwhile to explain in the methods section at least the structure of the tree.

Thirdly, I assume you handle uncertainty by sensitivity analysis. But how exactly did you do that? Was some form of boot-strapping involved? Or how did you get the distribution of variables?

In other words: If you can describe more precisely what you most likely did anyhow, I will support publishing without doubt.

Reviewer #2: It would be helpful if the autors state that the 444 Euros is taking into account the number of live births, not that this is the cost for an individual pregnancy child with ct. Otherwise everything clear and worthwhile for readers to know.

Reviewer #3: The paper entitled « Prevention of congenital toxoplasmosis in France using prenatal screening: A decision analytic economic model” by Sawers et al. describes a cost-benefit analysis of prenatal screening for T. gondii in France. The experience drawn from this disease in treated children in France, a country with historically successful management of congenital toxoplasmosis, supports the value of prenatal screening. The authors show it to be cost-effective, in addition to reducing or eliminating the devastating outcomes caused by the parasite. The paper is well written and interesting to read. Some points need to be addressed to help in the rigor of the presentation of this manuscript.

- It would be useful to compare the cost-benefit of prenatal screening in France to countries that have rejected both prenatal and neonatal screening (e.g., UK, Denmark, and Sweden). Someone would argue that the reasons for this relate to the clinical harms associated with false-positive diagnoses, low and uncertain benefits of treatment, harms associated with treatment, and high costs of prenatal screening.

- Line 334, the authors state, “The finding that maternal screening is cost saving in both Austria and France – and would be in the US if implemented there – serves to reinforce confidence in ongoing prenatal toxoplasmosis screening programs.” This statement is not supported by the authors investigation. Prenatal screening is not routine in the US. It is hard to be implemented because of the low prevalence of the disease, high cost of screening, high fee for service health care providers, and a high proportion of the uninsured population.

-The authors need to present evidence for the effectiveness of alternative public health interventions, when implemented with prenatal screening.

- It would be useful to define the demographic characteristics (e.g., race, ethnicity, education level) of the women participated in this analysis.

- Given the growing number of virulent Toxoplasma in central and south America, how could the costs and benefits of prenatal screening and public health interventions be prioritized in these countries according to the risks of acquiring infection during pregnancy, the timing of infection, and the capacity of health care systems?

- Rewrite the sentence of 322-323. The evidence for treatment is not sufficiently robust. Data from Wallon et al. (2013) showed symptoms in congenitally infected children even when the mothers were treated during pregnancy.

- Figure 1a, b has low resolution and hard to read

- Line 318 sentence is unclear

- Line 107, word check

Reviewer #4: Important topic but

1-important confusion concerning the choice of the vocabulary (this study can not be a cost-benefit analysis as it is presented)

2-impossible to read the figures

3-there is an important ambiguity concerning the goal of the article (is it for the US or French decison makers?). The title needs to be modified.

4-I do not understand the fact to have 2 trees and 1 result.

6. PLOS authors have the option to publish the peer review history of their article (what does this mean?). If published, this will include your full peer review and any attached files.

Reviewer #1: **Yes: **Steffen Flessa

Reviewer #2: No

Reviewer #3: No

Reviewer #4: No

---

## [Author Response · Author response to Decision Letter 0]

7 Apr 2022

Point-by-Point Responses to Reviewers: Rebuttal Letter

Response of authors to the Editor’s and reviewers’ queries and comments appear below in italics.

First, we want to thank you – the Editor and reviewers – for your help in improving our manuscript. We have responded to all of your comments. Your advice has helped us to submit a substantially improved manuscript to PLOS ONE. 

The Academic Editor says, “If applicable, we recommend that you deposit your laboratory protocols in protocols.io to enhance the reproducibility of your results.” 

Our Response: The research on which this paper is based is not an experiment conducted in a laboratory and so there are no laboratory protocols to deposit. This recommendation is not applicable to our submission.

Our submission meets PLOS ONE's style requirements, including those for file naming. https://journals.plos.org/plosone/s/file?id=wjVg/PLOSOne_formatting_sample_main_body.pdf and 

 Our Response: We have paid close attention to these formatting guidelines.

No authors have competing interests

Our Response: The authors have declared that no competing interests exist." We have already stated we have no competing interest. We are not sure what else you want us to do. We still have no competing interests.

In our original submission we noted we would provide repository information for our data at acceptance. Unfortunately, we have been confused by the words used here. The only repository we have is the Supporting Information Files attached to the article as separate files. Hence, there is no accession number or DOI. We have made changes to our Data Availability statement and describe these changes in cover letter in our resubmission. Our data are from publicly available sources and our calculations using those data are described in the article and its Supporting Information Files. 

4. Please amend your list of authors on the manuscript to ensure that each author is linked to an affiliation. Authors’ affiliations should reflect the institution where the work was done (if authors moved subsequently, you can also list the new affiliation stating “current affiliation” as necessary).

Our Response: The title page of the manuscript now follows the PLOS ONE style sheet.

Our Response: We have tried to follow these instructions. There are only two Supporting Information Files.

Reviewer's Responses to Questions and Comments to the Authors 

Our reactions to the editor’s and reviewers’ comments are in italics.

1. Is the manuscript technically sound, and do the data support the conclusions? 

Reviewer #1: Partly

Reviewer #2: Yes

Reviewer #3: Yes

Reviewer #4: Partly

Our Response: We believe that our manuscript is technically sound and that our conclusions are appropriately based on the data presented. As noted above, our manuscript does not report on a controlled experiment or a randomized controlled trial. Accordingly, we have no controls or sampling. However, our study is fully replicable. Indeed, our study replicates 2 earlier cost-benefit studies using a decision tree analysis of toxoplasmosis screening, both of which were published in PLOS NTDs.Our study design is replicable by others just as our paper replicates earlier studies.

 Two reviewers report that our paper only partly meets the criteria for success listed in the above paragraph.That judgement makes sense if Reviewers #1 and #4 expected our work to include a controlled experiment, sampling, and/or a randomized controlled trial, none of which belong in our manuscript. 

2. Has the statistical analysis been performed appropriately and rigorously?

Reviewer #1: Yes

Reviewer #2: Yes

Reviewer #3: Yes

Reviewer #4: Yes

3. Have the authors made all data underlying the findings in their manuscript fully available?

The PLOS Data Policy requires authors to make all data underlying the findings described in their manuscript fully available without restriction, with rare exception (please refer to the Data Availability Statement in the manuscript PDF file). The data should be provided as part of the manuscript or its supporting information, or deposited to a public repository. For example, in addition to summary statistics, the data points behind means, medians and variance measures should be available. If there are restrictions on publicly sharing data—e.g. participant privacy or use of data from a third party—those must be specified.

Reviewer #1: No

Reviewer #2: Yes

Reviewer #3: Yes

Reviewer #4: Yes

Our Response: The data used in producing our study is fully explained in the text of the article and in the Supplemental Information Files included in our submission. Is it possible that Reviewer #1 did not see the Supporting Information Files? (It appeared in the copy of the paper sent to reviewers on the very last page of the submission.) If he or she did not notice the Supplemental Information Files, then his or her response that underlying data in the manuscript are not fully available is no longer puzzling.

4. Is the manuscript presented in an intelligible fashion and written in standard English?

Reviewer #1: Yes

Reviewer #2: Yes

Reviewer #3: Yes

Reviewer #4: No

Our Response: Reviewer #4 notes that the manuscript we submitted was not intelligible and/or written in standard English. At any rate, without more specific criticisms, we have no way to react to this reviewer’s comment. We should point out that the lead author is a native English speaker who has published dozens of articles in distinguished academic journals and authored or edited five books issued by well-known publishing houses. We have carefully re-read the article and have corrected the few mistakes we found..

5. Review Comments to the Author

Reviewer #1 

This is an excellent paper. The topic is highly relevant, the methodology is appropriate, it is well-written. I support publishing under few conditions..

Firstly, your results strongly depend on the costs. Even if the tornado diagram (+/- 20 %) indicates that the intervention remains cost-saving, there is still a major question about the costs. I would like to know much more about how you calculated the costs. At least I would like to see a detailed analysis of the underlying literature. How did you calculate the intervention cost? What part of the life-long cost are direct and indirect? How were indirect cost / opportunity cost calculated? and if you take them from literature are they really comparable? What interest rate did you include? In other words: write much more about the costs. 

Our Response: The data used in producing our study are fully explained in the text of the article and in the Methodological Supplement (located in the Supporting Information File 1) included in our submission. The Supporting Information File has a six page single-spaced description of how we measured injury costs. The only way we can understand Reviewer #1’s response here is that he or she did not notice our Supporting Information Files on the last page of the original submission. For example, the reviewer asks “What interest rate did you include?” Nevertheless, several passages in the Supporting Information File 1 discuss different interest rates.

Secondly, I had severe problems to read the decision-tree. It might be worthwhile to explain in the methods section at least the structure of the tree.

Our Response: We have divided the image of the decision tree into 9 separate branches, each of which is easily read. These 9 files are now included in the Supplemental Information File 2. In the main text is an illegible version of the entire decision tree whose complexity is visually underscored.

Thirdly, I assume you handle uncertainty by sensitivity analysis. But how exactly did you do that? Was some form of bootstrapping involved? Or how did you get the distribution of variables? In other words: If you can describe more precisely what you most likely did anyhow, I will support publishing without doubt.

Our Response: A sensitivity analysis was performed using a tornado graph that compared the effects of changing the value of 24 explanatory variables by ± 20% on our output value. The explanatory variable with easily the most important impact on our output variable is the primary maternal infection prevalence. A second graph (Figure 3) shows that maternal screening is cost saving when the primary maternal infection rate is greater than .00055. These are our only sensitivity tests; no boot-strapping was utilized in our analysis. 

Reviewer #2

It would be helpful if the authors state that the 444 Euros is taking into account the number of live births, not that this is the cost for an individual pregnancy child with ct. Otherwise, everything is clear and worthwhile for readers to know. 

Our Response: Yes, the €444 was computed per child after taking into account all live births in France.

Reviewer #3 

The paper entitled “Prevention of congenital toxoplasmosis in France using prenatal screening: A decision analytic economic model” by Sawers et al. describes a cost-benefit analysis of prenatal screening for T. gondii in France. The experience drawn from this disease in treated children in France, a country with historically successful management of congenital toxoplasmosis, supports the value of prenatal screening. The authors show it to be cost-effective, in addition to reducing or eliminating the devastating outcomes caused by the parasite. The paper is well written and interesting to read. 

Some points need to be addressed to help in the rigor of the presentation of this manuscript.

It would be useful to compare the cost-benefit of prenatal screening in France to countries that have rejected both prenatal and neonatal screening (e.g.,). Some would argue that the reasons for this relate to the clinical harms associated with false-positive diagnoses, low and uncertain benefits of treatment, harms associated with treatment, and high costs of prenatal screening.

Our Response: The reviewer is raising an important point: to our knowledge, no standard medico- economic assessment in several countries has been published supporting the decision not to screen. It is true that screening tests might yield false negative or negative results and that drug prescription needs to respect rules. However in our experience, false positive tests results are less frequent than estimated and their impact can be strongly reduced by adequate diagnostic strategies and training, Moreover, we find treatment is well tolerated in mothers and their children. We would not otherwise prescribe them at such a large scale in France. We therefore call for using medico economic appraisal to make informed decision-based realistic estimates. We have added a sentence line 334 of the text. Ours is now the third cost-benefit analysis showing that the benefits of maternal screening exceed its cost. 

Line 334, the authors state, “The finding that maternal screening is cost saving in both Austria and France – and would be in the US if implemented there – serves to reinforce confidence in ongoing prenatal toxoplasmosis screening programs.” This statement is not supported by the authors’ investigation. Prenatal screening is not routine in the US. It is hard to be implemented because of the low prevalence of the disease, high cost of screening, high fee for service health care providers, and a high proportion of the uninsured population. 

Our Response: The reviewer is correct when explaining the difference between the US and other countries, but Stillwaggon’s et al.’s study shows that even with those differences, the potential benefits of prenatal screening would exceed by a substantial amount its cost. Research often tries to estimate what the future might bring, and that is what our study attempts to do. 

The authors need to present evidence for the effectiveness of alternative public health interventions, when implemented with prenatal screening. 

Our Response. Prenatal screening requires screening all pregnant women whose immunity status for Toxoplasma is unknown to identify those who are susceptible. This systematic identification of women at risk naturally leads to their education on how to avoid infection. This probably contribute to reducing the incidence of maternal infections and thus the number of severely infected children, (and might contribute to explaining the decreasing incidence in France,) but the effect size cannot be precisely estimated, in the absence of any ethical possibility to randomize the delivery of any preventive advice. Similarly, identifying women who seroconverted during delivery allows submitting their newborns systematically to extensive diagnostic work up, including those who show no symptoms. Treatment and follow up of all infected newborns probably contributes to their good overall functional prognosis. Education of women found to be non immunized is natural/ We could not do otherwise--it just goes together with screening. The same is true for screening newborns at birth--no one would even consider not doing it. 

Nevertheless, the specific contribution of each single intervention is impossible to estimate precisely. 

We would, however, point out that women who are informed of the risk of Toxoplasma infection might be likely to be more aware of other prevention measures. The higher incidence of listeriosis reported in a French area of higher prevalence for toxoplasmosis might be considered as an indirect evidence of these spillover benefits of screening.. 

It would be useful to define the demographic characteristics (e.g., race, ethnicity, education level) of the women participated in this analysis. 

Our Response: Our study is not a trial or experiment, and it is not a study of a sample of women. Our data seek to measure costs and benefits for all women in France. 

Given the growing number of virulent Toxoplasma in central and south America, how could the costs and benefits of prenatal screening and public health interventions be prioritized in these countries according to the risks of acquiring infection during pregnancy, the timing of infection, and the capacity of health care systems? 

Our Response: We would be delighted if someone would carry out the proposed project, but it is far beyond our resources. It would need to take into account the specific epidemiological characteristics in each setting; prevalence of toxoplasmosis in pregnant women, incidence of maternal infection, incidence of congenital toxoplasmosis and risk of long term lesions. At any rate, the benefits of screening are likely to correlate with the severity of the disease that is prevented. Accordingly, we invite local policy makers to set priorities. We add a sentence Line 331.

Rewrite the sentence of 322-323. The evidence for treatment is not sufficiently robust. Data from Wallon et al. (2013) showed symptoms in congenitally infected children even when the mothers were treated during pregnancy. 

Our Response: The “Wallon et al. study” provided evidence that starting treatment early reduces the risk and severity of congenital toxoplasmosis. The comparison did not compare treatment vs no treatment, but treatment started early vs. later. Screening pregnant women every month (to insure prompt detection and treatment) is a requisite for obtaining the full benefit of treatment. The efficiency of the French programs could certainly be improved by improving compliance with the monthly testing. 

These aggregated costs might have reduced the difference in costs and benefits between screening and no screening. That, however, was not found by Binquet et al. They found maternal screening for toxoplasmosis to be cost effective.

Our response: to respond adequately to this query we would need to know what are the aggregated costs mentioned by the reviewer 4.

Figure 1a, b has low resolution and hard to read

Our response: Any image that encompasses the entire decision tree and is also readable would be a meter or more in height, that is, would not fit on a single page. Consequently, we include within the text of the article a single image of the entire tree which is almost entirely illegible but it gives the reader a sense of the complexity of our statistical analysis. Then, in the Supporting Information File 2 we include 9 separate images. Each one is an image of one of the nine major branches of the tree. The numbers and words on each of those branches are easily readable. 

Line 318 sentence is unclear

Our response: Line 318 to line 324 acknowledges a simplification that we chose to make to exclude the possibility for acute maternal infection to be detected by clinical signs; these are present in 20 to 30% of cases in immunocompetent adults but are non-specific and often identified only after seroconversion is diagnosed.” We did this for the sake of allowing comparison with the previous CBA studies.

Line 107, word check

 Our response: We fixed the “T” in line 110.

Reviewer #4 

Important topic but

1-important confusion concerning the choice of the vocabulary (this study cannot be a cost-benefit analysis as it is presented)

Our Response: Two articles in a distinguished journal (Stillwaggon et al. and Prusa et al. in PLOS NTDs) have together been cited almost 200 times without receiving any criticism for describing our method as a cost-benefit analysis. Without your explanation for why you think our study is not a cost-benefit analysis we have no way to address your comment.

2-impossible to read the figures

Our Response. We agree. As noted above, we have reformatted the tree figures. We divided the main tree into 9 separate branches that are included in our Supporting Information File (S2). The figure of the entire tree is still illegible, but necessarily so: it is too big to fit on a single page. Nevertheless, each of the 9 sub-branches easily fit on a single page and are easily readable.

3-there is an important ambiguity concerning the goal of the article (is it for the US or French decision makers?). 

Our Response: The goal of our article is to illustrate how medico – economic evaluation can help make informed decisions regarding the pursuit of ongoing screening programs for congenital toxoplasmosis or the implementation of new universal programs. Whether or not universal screening is appropriate has been debated for a long time, but the decreasing prevalence of toxoplasmosis in pregnant women further reinforces the need to answer this question objectively. We used data routinely collected in the context of the most comprehensive screening program to date and the largest number of pregnant women screened daily. Our results will inform French (and both Austrian and Slovenian) decision-makers that maternal toxoplasmosis screening is cost saving and so maintaining existing screening programs makes economic sense. The same demonstration has been made for the US. But we also expect that decision makers elsewhere with an interest in toxoplasmosis will be interested in our results and encouraged to perform an appraisal such as ours. Our model is a generalizable tool for all policy makers worldwide and not only for the French or American ones. Standardization of tools is a key issue to allow.

The title needs to be modified.

Our Response: We hope that the explanation given above will convince the reviewer that the study is not about US, but about the cost and benefits of a screening program such as the one that is currently used in France. We have modified the text (see previous paragraph) rather than the title. We would however be pleased to hear your suggestions for a different title for our article. 

4-I do not understand the fact to have 2 trees and 1 result.

Our Response: We have only one tree with nine branches. Our methodology is to compare the two main branches, one of which measures the benefits of prenatal testing and treatment and the other with no prenatal testing and treatment. The main branching is where the tree divides between screening and no screening and the ratio of the two gives us the principal measure of the economic advantage of screening over no screening.

 6. PLOS authors have the option to publish the peer review history of their article. If published, this will include your full peer review and any attached files.

Our Response: We have no opinion about our peer review history.

Do you want your identity to be public for this peer review? For information about this choice, including consent withdrawal, please see our Privacy Policy.

Reviewer #1: Yes: Steffen Flessa

Reviewer #2: No

Reviewer #3: No

Reviewer #4: No

---

## [Decision Letter · Decision Letter 1]

20 Apr 2022

PONE-D-21-31899R1Prevention of congenital toxoplasmosis in France using prenatal screening: A decision-analytic economic model

PLOS ONE

Dear Dr. Sawers,

Thank you for submitting your manuscript to PLOS ONE. After careful consideration, we feel that it has merit but does not fully meet PLOS ONE’s publication criteria as it currently stands. Therefore, we invite you to submit a revised version of the manuscript that addresses the points raised during the review process.

You have addressed nearly all reviewer concerns and the manuscript is well-written. Only two issues remain:

1) I agree with reviewer #1 that additional information about the economic analysis assumptions should be added in the methods section of the main manuscript. This does not need to be extensive - a few sentences with the most relevant information will be helpful for readers with expertise in cost effectiveness analyses.

2) Figure 1 is too small to be legible. Please move it to supplemental materials with the larger snapshot images that are already in supplemental images.

We look forward to receiving your revised manuscript.

Kind regards,

Jodie Dionne-Odom, MD

Academic Editor

PLOS ONE

Journal Requirements:

Additional Editor Comments (if provided):

The authors worked hard to address all of the reviewer concerns and the manuscript is well-written. Only two issues remain.

1) I agree with reviewer #1 that additional information about the economic analysis assumptions should be added in the methods section of the main manuscript. This does not need to be extensive - a few sentences with the most relevant information will be helpful for readers with expertise in cost effectiveness analyses.

2) Figure 1 is too small to be legible. Please move it to supplemental materials with the larger snapshot images that are already in supplemental images.

Reviewers' comments:

Reviewer's Responses to Questions

**Comments to the Author**

1. If the authors have adequately addressed your comments raised in a previous round of review and you feel that this manuscript is now acceptable for publication, you may indicate that here to bypass the “Comments to the Author” section, enter your conflict of interest statement in the “Confidential to Editor” section, and submit your "Accept" recommendation.

Reviewer #1: (No Response)

Reviewer #3: All comments have been addressed

Reviewer #4: All comments have been addressed

2. Is the manuscript technically sound, and do the data support the conclusions?

Reviewer #1: Yes

Reviewer #3: Yes

Reviewer #4: Yes

3. Has the statistical analysis been performed appropriately and rigorously? 

Reviewer #1: Yes

Reviewer #3: Yes

Reviewer #4: Yes

4. Have the authors made all data underlying the findings in their manuscript fully available?

Reviewer #1: Yes

Reviewer #3: Yes

Reviewer #4: Yes

5. Is the manuscript presented in an intelligible fashion and written in standard English?

Reviewer #1: Yes

Reviewer #3: Yes

Reviewer #4: Yes

6. Review Comments to the Author

Reviewer #1: As I had commented already in the first review, this is a very good paper. My main concern was that the methodology is not clear. You entered lots of information into the Supporting Information Files. That was my main critique: it is in the wrong place. The methodology (e.g. interest rate) is so relevant that it MUST be in the main body of the text. This is standard in health economic papers. From my perspective, You cannot put most crucial elements of the methodology into an appendix or attachment.

However, I would leave this to the editors. If they think it is o.k., then it is fine with me.

Reviewer #3: (No Response)

Reviewer #4: (No Response)

7. PLOS authors have the option to publish the peer review history of their article (what does this mean?). If published, this will include your full peer review and any attached files.

Reviewer #1: **Yes: **Steffen Flessa

Reviewer #3: **Yes: **KAMAL EL BISSATI

Reviewer #4: No

---

## [Author Response · Author response to Decision Letter 1]

10 May 2022

We agreed to all of the comments made by the editor and reviewers and have responded appropriately by revising the text of the article.

---

## [Decision Letter · Decision Letter 2]

30 May 2022

PONE-D-21-31899R2Prevention of congenital toxoplasmosis in France using prenatal screening: A decision-analytic economic modelPLOS ONE

Dear Dr. Sawers,

Thank you for submitting your manuscript to PLOS ONE. After careful consideration, we feel that it has merit but does not fully meet PLOS ONE’s publication criteria as it currently stands. Therefore, we invite you to submit a revised version of the manuscript that addresses the points raised during the review process.

The Auhors have addressed to all the comments and the manuscript results to be deeply improved, however only few reccomendations by the Reviewer#3 could further improve it in order to be better clear to the reader.

We look forward to receiving your revised manuscript.

Kind regards,

Adriana Calderaro

Academic Editor

PLOS ONE

Journal Requirements:

Reviewers' comments:

Reviewer's Responses to Questions

**Comments to the Author**

1. If the authors have adequately addressed your comments raised in a previous round of review and you feel that this manuscript is now acceptable for publication, you may indicate that here to bypass the “Comments to the Author” section, enter your conflict of interest statement in the “Confidential to Editor” section, and submit your "Accept" recommendation.

Reviewer #1: All comments have been addressed

Reviewer #3: All comments have been addressed

Reviewer #4: All comments have been addressed

2. Is the manuscript technically sound, and do the data support the conclusions?

Reviewer #1: Yes

Reviewer #3: Yes

Reviewer #4: Yes

3. Has the statistical analysis been performed appropriately and rigorously? 

Reviewer #1: Yes

Reviewer #3: Yes

Reviewer #4: Yes

4. Have the authors made all data underlying the findings in their manuscript fully available?

Reviewer #1: Yes

Reviewer #3: Yes

Reviewer #4: Yes

5. Is the manuscript presented in an intelligible fashion and written in standard English?

Reviewer #1: Yes

Reviewer #3: Yes

Reviewer #4: Yes

6. Review Comments to the Author

Reviewer #1: (No Response)

Reviewer #3: (No Response)

Reviewer #4: The two comments have been fully addressed by the authors. Thank you very much.

I know how complicated it is to export decision trees from the software Tree Age , but I think that a global view (even short and summarized) of the decision tree is missing. It would guide the reader to add all the figures (1b to 1i) in an entire tree. Moreover, when I opened the Supporting Information file, some figures were on a "portrait" format instead of landscape". This could be changed. Moreover, the authors provided formulas to calculate costs at the end of each arm but without any legend to understand the abbreviation of the variables used.

7. PLOS authors have the option to publish the peer review history of their article (what does this mean?). If published, this will include your full peer review and any attached files.

Reviewer #1: **Yes: **Steffen Flessa

Reviewer #3: No

Reviewer #4: No

---

## [Author Response · Author response to Decision Letter 2]

13 Jul 2022

The editor requested three changes to our submitted manuscript. They were "I think that a global view (even short and summarized) of the decision tree is missing. It would guide the reader to add all the figures (1b to 1i) in an entire tree. Moreover, when I opened the Supporting Information file, some figures were on a "portrait" format instead of landscape". This could be changed. Moreover, the authors provided formulas to calculate costs at the end of each arm but without any legend to understand the abbreviation of the variables used." We have responded to all three requests for change in our submission by making the requested changes. Larry Sawers Corresponding Author

---

## [Decision Letter · Decision Letter 3]

16 Aug 2022

Prevention of congenital toxoplasmosis in France using prenatal screening: A decision-analytic economic model

PONE-D-21-31899R3

Dear Dr. Sawers,

We’re pleased to inform you that your manuscript has been judged scientifically suitable for publication and will be formally accepted for publication once it meets all outstanding technical requirements.

Kind regards,

Adriana Calderaro

Academic Editor

PLOS ONE

Additional Editor Comments (optional):

Reviewers' comments:

Reviewer's Responses to Questions

**Comments to the Author**

1. If the authors have adequately addressed your comments raised in a previous round of review and you feel that this manuscript is now acceptable for publication, you may indicate that here to bypass the “Comments to the Author” section, enter your conflict of interest statement in the “Confidential to Editor” section, and submit your "Accept" recommendation.

Reviewer #3: All comments have been addressed

2. Is the manuscript technically sound, and do the data support the conclusions?

Reviewer #3: Yes

3. Has the statistical analysis been performed appropriately and rigorously? 

Reviewer #3: Yes

4. Have the authors made all data underlying the findings in their manuscript fully available?

Reviewer #3: Yes

5. Is the manuscript presented in an intelligible fashion and written in standard English?

Reviewer #3: Yes

6. Review Comments to the Author

Reviewer #3: The authors have adequately addressed the reviewers' comments and I feel that this manuscript is now acceptable for publication.

7. PLOS authors have the option to publish the peer review history of their article (what does this mean?). If published, this will include your full peer review and any attached files.

Reviewer #3: **Yes: **KAMAL EL BISSATI
